# Analysis of S-Adenosylmethionine and S-Adenosylhomocysteine: Method Optimisation and Profiling in Healthy Adults upon Short-Term Dietary Intervention

**DOI:** 10.3390/metabo12050373

**Published:** 2022-04-20

**Authors:** Aida Corrillero Bravo, Maria Nieves Ligero Aguilera, Nahuel R. Marziali, Lennart Moritz, Victoria Wingert, Katharina Klotz, Anke Schumann, Sarah C. Grünert, Ute Spiekerkoetter, Urs Berger, Ann-Kathrin Lederer, Roman Huber, Luciana Hannibal

**Affiliations:** 1Laboratory of Clinical Biochemistry and Metabolism, Department of General Pediatrics, Adolescent Medicine and Neonatology, Medical Center—University of Freiburg, Faculty of Medicine, University of Freiburg, 79106 Freiburg, Germany; aidacorrillero.ac28@gmail.com (A.C.B.); nievesligero98@gmail.com (M.N.L.A.); nahuelriosm@gmail.com (N.R.M.); lennart.moritz@uniklinik-freiburg.de (L.M.); wingertvictoria@gmail.com (V.W.); katharina.klotz@uniklinik-freiburg.de (K.K.); anke.schumann@uniklinik-freiburg.de (A.S.); urs.berger@uniklinik-freiburg.de (U.B.); 2Department of General Pediatrics, Adolescent Medicine and Neonatology, Medical Center—University of Freiburg, Faculty of Medicine, University of Freiburg, 79106 Freiburg, Germany; sarah.gruenert@uniklinik-freiburg.de (S.C.G.); ute.spiekerkoetter@uniklinik-freiburg.de (U.S.); 3Center for Complementary Medicine, Department of Internal Medicine II, Medical Center—University of Freiburg, Faculty of Medicine, University of Freiburg, 79106 Freiburg, Germany; ann-kathrin.lederer@uniklinik-freiburg.de (A.-K.L.); roman.huber@uniklinik-freiburg.de (R.H.)

**Keywords:** S-adenosylmethionine, S-adenosylhomocysteine, methionine, cobalamin, folate, homocysteine, targeted metabolomics, inborn errors of metabolism

## Abstract

S-adenosylmethionine (SAM) is essential for methyl transfer reactions. All SAM is produced de novo via the methionine cycle. The demethylation of SAM produces S-adenosylhomocysteine (SAH), an inhibitor of methyltransferases and the precursor of homocysteine (Hcy). The measurement of SAM and SAH in plasma has value in the diagnosis of inborn errors of metabolism (IEM) and in research to assess methyl group homeostasis. The determination of SAM and SAH is complicated by the instability of SAM under neutral and alkaline conditions and the naturally low concentration of both SAM and SAH in plasma (nM range). Herein, we describe an optimised LC-MS/MS method for the determination of SAM and SAH in plasma, urine, and cells. The method is based on isotopic dilution and employs 20 µL of plasma or urine, or 500,000 cells, and has an instrumental running time of 5 min. The reference ranges for plasma SAM and SAH in a cohort of 33 healthy individuals (age: 19–60 years old; mean ± 2 SD) were 120 ± 36 nM and 21.5 ± 6.5 nM, respectively, in accordance with independent studies and diagnostic determinations. The method detected abnormal concentrations of SAM and SAH in patients with inborn errors of methyl group metabolism. Plasma and urinary SAM and SAH concentrations were determined for the first time in a randomised controlled trial of 53 healthy adult omnivores (age: 18–60 years old), before and after a 4 week intervention with a vegan or meat-rich diet, and revealed preserved variations of both metabolites and the SAM/SAH index.

## 1. Introduction

S-adenosylmethionine (SAM) is an important metabolic intermediate and the cell’s universal methyl donor for the synthesis and modification of various biomolecules, such as nucleic acids, histones, proteins, hormones, phospholipids, and biogenic amines [1]. SAM is synthesised de novo from the essential amino acid methionine (Met) by the addition of an adenosyl group into methionine via methionine adenosylmethyltransferase (MAT) (Figure 1a). The demethylation of SAM produces S-adenosylhomocysteine (SAH). The ratio between SAM and SAH is commonly referred to as the “methylation index” that reflects the methylation capacity of a cell or an organism [1]. SAM-dependent enzymes catalyse the transfer of methyl groups into acceptor molecules via a variety of mechanisms that continue to be investigated extensively [2,3,4,5]. At high concentrations, SAH competitively inhibits SAM-dependent methyl transferases, and thus methylation is controlled through product inhibition and depends on the efficient hydrolytic conversion of SAH into the downstream metabolite homocysteine (Hcy) (Figure 1a). Homocysteine is a crucial metabolite that connects the methionine and folate cycles and the transsulfuration pathway, and whose partition between these pathways is carefully regulated [6]. The remethylation of Hcy to form Met depends on methionine synthase, an enzyme that requires cobalamin (vitamin B_12_) and 5-methyltetrahydrofolate (vitamin B_9_) as cofactors [7]. The conversion of Hcy to cystathionine and cysteine is carried out by cystathionine β-synthase (CBS) and cystathione γ-lyase (CSE), respectively, which utilise pyridoxal phosphate (vitamin B_6_) as a cofactor. SAM is an allosteric activator of CBS [8] and an allosteric inhibitor of methylentetrahydrofolate reductase (MTHFR) [9]. The enzyme MTHFR catalyzes the conversion of 5,10-methylentetrahydrofolate to 5-methyltetrahdrofolate. While SAM is recognised as the universal methyl donor in all living cells [10], this small molecule is also a source of other groups, such as amino, aminoalkyl, and ribosyl moieties, which have been reviewed elsewhere [11]. 

Interest in the accurate determination of SAM and SAH has emerged from their biomarker value for the diagnosis of a number of inborn errors of metabolism (IEM) that disrupt folate, cobalamin, and methionine metabolism [12,13,14,15]. The concentrations of SAM and SAH in plasma were reported to vary also in certain IEM unrelated to the abovementioned pathways [16], as well as with certain dietary patterns [17,18,19], chronic diseases [20,21,22,23], and sepsis [24]. The monitoring of SAM metabolism appears of relevance in the context of the proposed use of oral SAM for the treatment of certain neurophyschiatric disorders [25]. While structurally similar to SAH (Figure 1b), SAM is unstable under neutral and alkaline conditions, which requires careful sample handling, storage, and transport if intended for diagnostic purposes [26]. This, along with the naturally low concentration of these metabolites in plasma, has led to sustained efforts toward the development and optimisation of methods for diagnostic purposes. 

Herein, we describe an LC-MS/MS method for the determination of SAM and SAH in plasma, urine, cells, and conditioned culture medium. A side-by-side comparison of this method versus a method currently in place for diagnostic purposes showed agreement and correlation of SAM and SAH determinations in human plasma from patients with IEM that disrupt SAM and SAH homeostasis. As part of the methionine cycle, SAM synthesis and metabolism are dependent on the consumption of essential nutrients, such as methionine, folic acid, and cobalamin, whose intakes vary between diets. A growing number of people are transitioning toward plant-based diets worldwide [27,28,29,30,31]. Whether their motivations include mitigation of climate change and food insecurity, animal welfare, health improvement, or other, the study of how nutrition impacts metabolism in such dieters has gained renewed interest, both in adult and paediatric populations [32,33,34,35]. We hypothesized that SAM and SAH concentration in human plasma may vary upon transition from an omnivorous diet into a vegan or meat-rich diet. To test the effect of a plant-based diet on SAM and SAH metabolism, we examined SAM and SAH concentrations in the plasma and urine of a well-characterised cohort of healthy omnivorous adults participating in a randomised controlled trial (RCT) with an intervention with either a vegan or a meat-rich diet (VD and MD, respectively) for 4 weeks [36,37,38,39]. Our study revealed similar variations in SAM and SAH concentrations from baseline to end of trial in the plasma and urine of healthy adults, regardless of dietary pattern. 

## 2. Results

### 2.1. Separation and Quantification of SAM and SAH by LC-MS/MS

The concentrations of SAM, SAH, and Crea from solutions prepared as described in the methods (Figure 2a) could be determined simultaneously by reverse-phased LC using isocratic solvent conditions and MS as a detection system. Chromatograms of analyte and isotopically labelled internal standards are shown in Figure 2b. Retention times, optimised conditions, and mass transitions for SAM and SAH are provided in Table 1. The parameters for the detection of Crea and D_3_-Crea are as described in a previous publication [40].

### 2.2. Linearity Range and Limit of Quantification

The assay showed a suitable linearity of calibration curves from 0 to 1 µM of SAM and SAH (Figure 2c), which were extendable to 10 µM (Appendix A). This expanded linearity range permitted the quantification of SAM and SAH in undiluted plasma from patients exhibiting SAM > 1000 nM. Under our experimental conditions, the limits of detection (LOD) for SAM and SAH were 5 nM and 1 nM, respectively. The limits of quantification were 10 nM and 3 nM for SAM and SAH, respectively.

### 2.3. Recovery, Carry Over, and Analysis of Matrix Effect

The recovery of SAM and SAH from plasma was approximately 50%, as examined by spiking plasma samples with D_3_-SAM and ^13^C_5_-SAH. This was lower than the recovery of the diagnostic method for the determination of plasma SAM and SAH in place in our clinic and elsewhere [41,42], which are approximately 75% and 65–80%, respectively. The comparatively higher recovery of SAM and SAH in those methods may be related to gains from larger sample volume and concentration via solid-phase extraction, which is a step excluded in our method. Since the lower recovery of extraction did not significantly impact the trueness and precision of SAM and SAH determinations, we decided to maintain the one-step acidic methanol extraction protocol. Carry over was determined by water injections applied right after sample runs. No significant carry over was observed under our experimental conditions. SAM and SAH exhibited marked matrix effects in plasma and urine (Figure 3), as well as in cell lysates and culture medium (Figure 4). Because plasma and urine are the main specimens used for diagnostic purposes, we examined the matrix effects by two independent users in our laboratories (Figure 3a–d). The fact that the two matrices impact SAM and SAH detection differently confirms the absolute requirement of isotopic dilution for quantification under our experimental conditions. The components of cell lysates and conditioned culture medium also impact SAM and SAH, albeit comparably less markedly than as seen for plasma (Figure 4). While the components of the cell lysate had minor effects on the signal intensities of SAM and SAH, mild ion enhancement was observed for SAH in culture medium. Based on the results of the matrix effects for the different specimens and the response of analyte peak areas with varying injection volumes (Appendix A), measurements were carried out with an injection volume of 5 µL.

### 2.4. Determination of SAM and SAH in the Plasma of Healthy Adults

To compare the performance of our method with previously reported data, the total concentrations of SAM and SAH were determined in the plasma samples collected from adult healthy individuals. The cohort included 33 individuals (10 males and 23 females) between 19 and 60 years old (age distribution provided in Appendix A). The concentrations of plasma SAM and SAH were found to be 120.6 ± 18.1 nM and 21.5 ± 3.2 nM, respectively (Table 2). These concentrations and the corresponding SAM/SAH ratios agree with the reference values reported in other studies (Table 4 in next subsections and Appendix A). Creatinine was also examined for comparison as an index stable metabolite in plasma [43].

### 2.5. Determination of SAM and SAH in the Plasma of Patients with Inborn Errors of Metabolism

As a further step to test the performance of SAM and SAH determination, we compared SAM and SAH concentrations in identical plasma samples measured with this method versus a different method in the diagnostic laboratory at our Children’s Hospital, which is based on a published procedure [42]. The samples were from patients with suspected or confirmed IEM and were received in our research laboratory in anonymous and blind forms. Independently determined SAM and SAH concentrations were compared via Bland–Altman plots (Figure 5). The results from this assessment show agreement and a correlation of SAM and SAH concentration between the two methods. 

### 2.6. Profile of SAM and SAH in a Randomised Control Trial in Healthy Subjects with Dietary Intervention with a Vegan (VD) or a Meat-Rich (MD) Diet

#### 2.6.1. Concentration of SAM, SAH, and Creatinine in the Plasma and Urine of Healthy Omnivores at the Start of the Trial

We first examined the concentrations of SAM, SAH, and creatinine prior to initiation of the interventional dietary trial (Table 3). At baseline, all subjects adhered to a balanced omnivorous diet [37] and exhibited plasma concentrations of SAM, SAH, and creatinine within the reference ranges and cut-offs reported for adult healthy subjects by other groups (Table 4 and Table 5).

#### 2.6.2. Effect of Dietary Intervention on the Concentration of SAM and SAH in the Plasma and Urine of Healthy Subjects

To assess whether transitions from omnivorous to plant-based (VD) or meat-rich (MD) diets influence systemic (plasma) and excreted (urine) concentrations of SAM and SAH, these metabolites were measured at the start of the trial and after a 4 week dietary intervention. Table 6 shows the results of the statistical analysis for the interaction of fixed effects diet (VD and MD) and time (baseline and end). No statistically significant interactions were identified upon the respective dietary transitions over time (Table 6). Both plasmatic (Figure 6) and urinary (Figure 7) concentrations of SAM and SAH, as well as the SAM/SAH ratios, varied similarly over the course of 4 weeks, irrespective of diets. Plasma SAM was slightly higher at the end of the trial compared to baseline, both for individuals on VD and MD, and yet these variations fall within the reference range for healthy individuals. The urinary excretion of SAM and SAH increased slightly over time for both randomised diet groups, again within the reference range of healthy individuals.

#### 2.6.3. Associations of Plasma and Urinary SAM and SAH before and after Dietary Intervention

The metabolite concentrations and ratios determined in this study were subjected to correlation analysis in an attempt to identify differences between baseline conditions at the start of the trial (omnivores) and after dietary intervention (omnivores after VD or after MD). The results of regression analysis are shown in Figure 8 and Table 7. Regression plots of plasma versus urinary concentrations of SAM and SAH, as well as the SAM/SAH index, show substantial dispersion of data points (the outliers removed for this analysis are shown in Appendix A). Plasma and urinary SAH exhibited an association, which was, however, not influenced by diet or time under the conditions of our study. The significant factor of time observed for SAM and SAM/SAH is in line with results of GLMM, as presented in Table 6. Considering the marked dispersion of concentrations in these biological compartments, and the presence of outliers that were herein excluded (Appendix A), a larger dataset is required to test whether these findings represent a biologically relevant variation. This correlation analysis confirms that the relationship between plasma and urinary concentrations of SAM and SAH is essentially unmodified by the dietary intervention investigated herein.

## 3. Discussion

The primary objective of this work was to optimise the quantification of SAM and SAH in biological samples. We determined reference ranges in a cohort of adult healthy controls, validated assay reliability by comparison with an existing LC-MS/MS method utilised for diagnostic purposes, and investigated the effect of diet on plasmatic and urinary concentrations of these metabolites.

The optimisation of an LC-MS/MS method was undertaken for the determination of SAM and SAH. The concentration of metabolites in biological specimens is sensitive to a variety of pre-analytical factors that include fasting status of participating subjects, sample transportation delays, storage temperature, use of additives, and repeated freeze/thawing [53]. Efforts toward the optimisation of SAM and SAH detection and quantification have been made over the course of several years, as summarised in Table 4 (plasma) and Table 5 (urine). These methods have all proven reliable for the determination of SAM and SAH. Our assay is based on isotopic dilution to account for matrix effects, it requires only 20 µL of sample (plasma, urine, culture medium, or cell lysate), does not require time-consuming and expensive solid-phase extraction, is completed with an LC-MS/MS running time of 5 min per sample, and employs reagents and solvents available in standard laboratories. Under our experimental conditions, the assay exhibited linearity over a broad range of concentrations (0.010-5 µM for SAM and 0.003-5 µM for SAH), which permits the analysis of SAM and SAH in undiluted plasma from healthy control subjects, as well as patients with disrupted SAM/SAH metabolism, in a single batch. In agreement with the existing literature [26], we observed that the stabilisation of SAM by acidification of plasma samples reduces its decomposition, enabling sample re-measurement after freeze-thawing and long-term storage. This is of extreme importance if sample preparation protocols are intended for diagnostic purposes. Lack of plasma preservation prior to workup results in measured concentrations of SAM and SAH that deviate from the reference ranges reported for adult healthy human individuals [54].

The concentration of SAM and SAH in plasma is modified by diet, liver disease, and inborn errors of metabolism that impair the formation of precursor metabolites or their degradation [15,17,55]. Our SAM and SAH determination in plasma from a cohort of healthy adults agrees well with the values reported in at least nine other studies (Table 4): SAM: 120 ± 18 nM, SAH: 21.5 ± 3.3 nM, and SAM/SAH: 5.6 ± 1.0.

Next, we investigated assay reliability in terms of agreement and correlation by generating Bland–Altman plots of identical IEM patient samples examined by a diagnostic method based on solid-phase extraction clean-up and LC-MS/MS versus our method employing acidic methanol extraction followed by LC-MS/MS. These samples included a broad range of low and high SAM and SAH concentrations in plasma, as typically seen in pediatric and adolescent patients with genetic disturbances in enzymes that control SAM and SAH homeostasis. The method showed reliability in classifying healthy versus ill, which supports its application both for diagnostic and research purposes. In particular, this method is compatible with existing sulfur-containing metabolite profiling [40] and is likewise transferrable to automated metabolomic platforms.

We undertook an analysis of SAM and SAH in adult healthy subjects transitioning from an omnivore to either a vegan or a meat-rich diet. An increasing number of individuals are transitioning toward plant-based dietary patterns worldwide [27,28,29,30,31]. Regardless of personal motivations, which may include mitigation of climate change and food insecurity, animal welfare, and health improvement, the study of metabolism and disease prevention in such dieters has gained growing interest, especially in pediatric and adolescent populations [32,33,34,35]. We employed our method for the assessment of SAM and SAH concentrations in a well-characterised cohort of healthy adults [36,37,38,39]. Of note, samples of plasma in this cohort were isolated and immediately frozen at −80 °C, without acidification prior to freezing. We applied the acidification protocol to plasma samples on the first cycle of freeze-thawing. SAM and SAH concentrations were determined in plasma and urine at the start (baseline) and after 4 weeks on the respective diets (end). For the first time, the results from our study show that both plasmatic and urinary concentrations of SAM and SAH, as well as the SAM/SAH index, remained within reported reference ranges for healthy subjects after a 4-week intervention with a plant-based diet. Correlation analysis showed that the dietary interventions tested in this RCT were without effect on the relationship between systemic and excreted concentrations of these metabolites. Importantly, both diets were isocaloric and nutrient-replete, with cobalamin being the only micronutrient available in lower amounts via intake in subjects randomised to the VD. While our previous study demonstrated a statistically significant reduction in the dietary intake of cobalamin (vitamin B_12_), as well as a reduced concentration of plasma holo-transcobalamin (holo-TC) [37] in the VD group, both B_12_ and holo-TC concentrations diminished within the reference range reported for healthy subjects at the end of the 4-week trial. Such reductions in the concentration of SAM and SAH within the boundaries of reference ranges are, by definition, unlikely to impact health. The other important vitamin modifier of SAM and SAH, folate, also was within reference ranges for healthy subjects both in terms of intake and total plasma concentration for both the VD and the MD [37]. Intake and plasma concentrations of methionine, the precursor of SAM, remained within reference ranges in both the VD and the MD [37]. The intake of Met is lower in individuals adhering to plant-based diets compared to diets containing proteins of animal origin [1]. In our trial, the predicted lower intake of Met in the VD group did not translate into reduced plasma Met [37], which we presume contributes to maintenance of SAM and SAH homeostasis. An alternative explanation for our findings is that the dietary intervention was too short in length to elicit measurable changes in the concentrations of SAM and SAH. Thus, the impact of nutrition on metabolism requires careful studying, as not all metabolites may be equally responsive to changes in nutrient intake. In addition, distinct compensatory pathways may be elicited to maintain homeostatic concentrations of essential metabolites such as SAM and SAH. A study that profiled energy metabolism in omnivorous, vegetarian, and vegan children showed distinct metabolite profiles possibly representing “diet-submetabolomes” in these populations [33]. The authors concluded that pediatric and adult studies on plant-based nutrition cannot be extrapolated to interpret one another [33]. Expanded metabolite profiling of the adult cohort investigated in this study (short-term intervention with a plant-based diet), as well as profiling in long-term adult omnivores, vegetarians, and vegans, is currently underway to comprehend the impact of nutrition on energy metabolism and on housekeeping metabolites.

### Strengths and Limitations of the Study

While other methods exist for the determination of SAM and SAH, ours has the advantage of utilizing very small sample quantities (20 µL of plasma, urine, or culture medium, and 500,000 cells), it does not require costly and laborious solid-phase extraction during sample preparation, it has an instrumental running time of only 5 min per sample, and it can be directly transferred to automated metabolomics platforms. The accurate assessment of SAM and SAH concentration in plasma requires freshly isolated plasma and rapid freezing, and, ideally, the acidification of samples prior to long-term storage and measurement. The instability of SAM at neutral and alkaline pH limits its inclusion in metabolomic platforms that run under such conditions of pH. Urine samples that exhibit alkaline pH due to pathological conditions would require acidification prior to freezing, as described for plasma. The absence of externally validated quality controls, such as those provided by the European Research Network for the evaluation and improvement of screening, the diagnosis and treatment of inherited disorders of metabolism (ERNDIM), or the National Institute of Standards and Technology (NIST) for other biomarkers, requires additional efforts in maintaining in-house quality controls to monitor assay performance. Despite these limitations, the method described herein permits the quantification of SAM and SAH in plasma, urine, cells, and cell culture medium, thereby supporting both diagnostic and research applications.

## 4. Materials and Methods

### 4.1. Specimen Collection, Pre-Analytics, and Storage

Plasma was obtained by the centrifugation of EDTA-blood at 9447x g for 1–3 min at room temperature. The plasma (0.5 mL) was transferred into safe-lock 1.5 mL Eppendorf tubes containing 50 µL of 1 M acetic acid and mixed by inversion according to published recommendations [26]. The acidified sample was stored at −80 °C until further use. Samples submitted from external partners, for which acidification prior to storage was not possible, were transported in dry ice and stored at −80 °C until the day of analysis. Acidification was applied on the first thawing event. Spontaneous urine was collected and frozen at −80 °C until further analysis, without acidification (the pH of urine in healthy human subjects is around 6 [56,57]). On the day of the experiment, urine samples were thawed, vortexed, and diluted 1:25 with water prior to sample preparation (see Section 4.5) for SAM, SAH, and creatinine determination. Human renal epithelial cells from proximal tubules and their corresponding conditioned culture media utilised in this study were grown, harvested, and stored as described in our previous work [58].

### 4.2. Cell Lysis and Protein Quantification

Cell pellets were lysed with ice-cold PBS supplemented with a 1% protease inhibitor cocktail as described [58]. Cells were disrupted by freeze-thawing and homogenisation with a cordless pestle motor and disposable pellet mixers (VWR Nr. 47747-366, Radnor, PA, USA). Intra- and extra-cellular metabolite concentrations were normalised by total protein concentration of the cell lysate and expressed as nmol metabolite/mg protein. Total protein concentration was determined using the bicinchoninic acid assay (Thermofisher Nr. 23225).

### 4.3. Preparation of Stock Solutions, Internal Standard (IS) Solution, and Extraction Solution

Stock solutions of SAM, SAH, and their identical isotopes were prepared in 0.1% formic acid in H_2_O and stored in aliquots at −80 °C. These solutions were stable for at least 12 months. Stock solutions of creatinine and D_3_-Crea were prepared in water and stored in aliquots at −80 °C. The internal standard solution mixture contained 1 µM D_3_-SAM, 1 µM ^13^C_5_-SAH, and 50 µM D_3_-creatinine prepared in 0.1% formic acid in H_2_O. The extraction solution consisted of 0.1% formic acid in methanol.

### 4.4. Preparation of Calibration Curves

Calibration curves consisted of 12 concentrations obtained by th serial dilution of the calibrator with the highest concentration (calibrator 1, 5 µM SAM and SAH, and 500 µM Crea). Briefly, 20 µL of each calibrator (1 to 12) were mixed by vortexing with 20 µL of DTT 0.5 M and 20 µL of IS and then incubated for 10 min at room temperature. 100 µL of extraction solution was added to the mixture and vortexed. Calibrator extracts were then transferred to HPLC vials for LC-MS measurement or frozen at −80 °C (SAM and SAH are stable for at least 2 weeks).

### 4.5. Preparation of Samples and Normalisation

For the analysis of SAM, SAH, and creatinine, 20 µL of plasma, urine (previously diluted 1:25 with H_2_O), or cell lysate was pipetted into 1.5 mL Eppendorf tubes. 20 µL of DTT 0.5 M and 20 µL IS were added to all samples and mixed by vortexing, followed by a 10 min incubation at room temperature. 100 µL of extraction solution was added to the mixture and vortexed. Samples were then centrifuged at 9447× *g* for 10 min at room temperature. The supernatant was transferred to HPLC vials for LC-MS measurement or frozen at −80 °C (samples are stable for at least 2 weeks). The concentrations of urinary SAM and SAH were normalised by the concentration of creatinine. The concentrations of SAM and SAH in cells and culture media were normalised by protein concentration.

### 4.6. Preparation of Quality Controls (QCs)

Externally validated QCs are unavailable for SAM and SAH. Quality control solutions were prepared in-house to contain 20, 60, and 120 nM of each metabolite. The trueness and precision were evaluated within and between experiments.

### 4.7. LC-MS/MS Method, Signal Processing and Quantification, and Validation

The separation of the analytes was performed on a Sunfire C8 column (3.5 µm, 4.6 × 100 mm) with a flow rate of 0.75 mL/min. An isocratic elution with 0.1% formic acid in methanol/water (5/95) for 5 min was employed. An injection volume of 5 µL was used. The analytes were optimally ionised in the positive mode. Detection was performed on a Sciex 6500+ ESI-tripleQ MS/MS (AB Sciex Germany GmbH, Darmstadt, Germany) on low mass mode (0–1000 Da) with instrument settings as described in our previous work [40]. Validation experiments utilised a diagnostic LC-MS/MS method that was published elsewhere [42]. Quantification of metabolites wascarried out with Analyst^®^ 1.6.3 software, 2015 AB Sciex. This method was optimised in accordance with guidelines of the European Medicines Agency for the validation of analytical procedures [59].

### 4.8. Matrix Effects

The target metabolites SAM and SAH are known to undergo significant ion suppression or enhancement in the presence of biological matrices, and hence the recommended use of solid-phase extraction for sample clean-up and isotopic dilution was the method of choice [26]. Matrix effects were evaluated and confirmed under the precise experimental conditions of our study, according to common practice recommendations [60,61,62]. Briefly, the effect of the matrices on the intensity of the studied ions was evaluated by comparing their peak areas in matrix-free solvent (0.1% formic acid in water, no matrix), matrix (plasma, urine, or cell lysate), and extracted matrix (plasma, urine, or cell lysate after extraction with 0.1% formic acid in methanol), with 1 µM D_3_-SAM, 1 µM ^13^C_5_-SAH, and 50 µM D_3_-creatinine. The response of analyte peak area with respect to injection volume was also examined (Appendix A).

### 4.9. Ethical Approval

Informed consent was obtained for all subjects included in this study under the ethical approvals EK-Nr. 218/17 and EK-Nr. 198/18, Freiburg, Germany, and clinical trial Nr. DRKS00011963; German Clinical Trial Registry.

### 4.10. Cohort Characteristics of the Randomised Controlled trial (RCT) with Intervention with Vegan (VD) or Meat-Rich (MD) Diets

Details on cohort characteristics and course of study can be found in our previous publication [37]. Briefly, participants were recruited between April and June of 2017 as part of a monocentric, randomised controlled trial. The study was approved by the ethical commission of the Medical Center, University of Freiburg (EK Freiburg 38/17), and registered at the German Clinical Trial Register (DRKS00011963). Criteria of eligibility comprised individuals aged between 18 and 60 years old with a body mass index (BMI) of between 21 and 30 kg/m^2^. Criteria of exclusion consisted of intake of medication (except for iodine), pregnancy or lactation, allergies, pre-existing vegetarian or vegan diets, eating disorders, aversion to vegan diet or animal products, smoking, and self-reported abuse of drugs or alcohol (>20 g/day). Participants had to have good command of the German language and willingness to complete a standardised nutritional protocol. The study included a run-in phase wherein individuals were trained extensively to maintain a balanced omnivore diet, according to the guidelines of the German Nutrition Association (DGE) [63], for one week. The target was maintenance of BMI. After this 1 week run-in phase, samples (fasted blood and urine) were taken and herein referred to as “Baseline”. The participants were then randomised into a meat-rich diet (MD, >150 g of meat per day) or a strict vegan diet (VD) for 4 weeks, herein referred to as “End”. Subjects received extensive training on their assigned diets, such that both diets remained isocaloric. A total of 53 individuals (20 males and 33 females) completed the run-in phase and underwent randomisation (26 participants allocated to VD and 27 participants allocated to MD). Of relevance to the present study, protein intakes at baseline were 86.9 ± 33.4 g and 94.0 ± 41.3 g for VD and MD (*p*-value = 0.474), respectively [37]. At the end of the trial, protein intakes were 70.5 ± 28.5 g and 112.4 ± 44.4 g for VD and MD, respectively (*p*-value < 0.001) [37]. Greater protein intake correlated with greater Met intake, which could influence SAM and SAH concentrations. During the intervention trial, participants had the opportunity to eat for free at the restaurant of the University Hospital Freiburg, where the vegan meal offered was especially designed for this study, in addition to the regular, mostly meat-based buffet. After 4 weeks, fasting blood and urine were collected and stored as described above. Participants remained in the study only if they showed a complete nutritional protocol, exhibited a stable body weight at the end of trial (compared to baseline), and had no further changes to their lifestyle.

### 4.11. Statistical Analysis

Metabolite data from the RCT were fitted to a generalised linear mixed model (GLMM) with distributions fitted specifically for each metabolite (SAM in plasma, SAM/SAH in plasma, SAM/Creatinine in urine, and SAH/Creatinine in urine followed a gamma distribution; SAM/SAH in urine followed a log-normal distribution; SAH in plasma and Creatinine in plasma followed a normal distribution). The fixed effects weredefined as diet (two levels, VD and MD) and time (two levels, baseline and end). Subjects were used as random effects. Fixed effects and their interactions were investigated using an analysis of deviance based on the mixed linear model. The significance level was set to α = 0.05.

Correlation analysis and Bland–Altman plots were used to assess the agreement between metabolite values obtained using the proposed method and the standard diagnostic method. Multiple linear regression analysis was performed to investigate associations between metabolites in plasma and urine. In this analysis, metabolites in urine were modeled as predictors of metabolites in plasma with the additional predictors of diet and time. Four models differing in their interaction terms were evaluated to find the best fitting for our data (see Appendix A). The selected model based on the Akaike information criterion (AIC) and the Bayesian information criterion (BIC) was the one with a full three-way interaction term. Prior to this analysis, data points falling above the 75th or below the 25th percentiles by a factor of three times the interquartile range were considered outliers, and hence they were removed from the correlation analysis. The statistical analysis was performed using R [64].

### 4.12. Data Visualisation

Graphical analysis was performed with GraphPad Prism 9 (Version 9.3.0 (345), 11 November 2021) and R [64].

## 5. Conclusions

Our LC-MS/MS platform permitted the fully quantitative determination of SAM and SAH in plasma, urine, culture medium, and cells. Validation in a cohort of healthy adults, as well as in a subset of patients with inborn errors of metabolism, showed a correlation and agreement in accordance with the reference ranges by specialised IEM diagnostic laboratories and the published works from different research groups (Table 5 and Table 6). Our study revealed that the plasma and urinary pools of SAM and SAH are overall preserved, remaining within the reference ranges for healthy adult individuals even upon intervention with a diet free of animal products (vegan diet). Further work is necessary to investigate the long-term effects (beyond 4 weeks) of this type of dietary intervention and, specifically, whether chronic adherence to plant-based nutrition modifies the homeostatic concentrations of certain metabolites.

## Figures and Tables

**Figure 1 metabolites-12-00373-f001:**
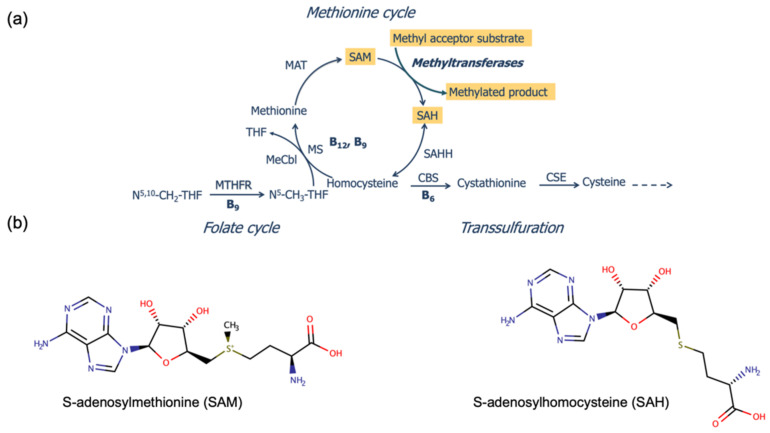
Metabolic pathways and structure of SAM and SAH. (**a**) All SAM and SAH derive from the methionine cycle, which relies on the availability of the water-soluble vitamins B_6_ (pyridoxal phosphate), B_9_ (folate), and B_12_ (cobalamin) that serve as enzyme cofactors. The methylation of methionine by methionine adenosyl transferase (MAT) produces SAM. In reactions catalysed by methyltransferases, SAM transfers methyl groups to a variety of acceptors that include nucleic acids, proteins, and neurotransmitter precursors. The demethylation of SAM yields SAH, which undergoes reversible conversion to homocysteine by SAH hydrolase (SAHH). Homocysteine can undergo remethylation via methionine synthase (MS), an enzyme that utilises both B_9_ (specifically, *N*^5^-methyltetrahydrofolate, synthesised by 5,10-methylentetrahydrofolate reductase, MTHFR) and B_12_ as cofactors, or by entering the transsulfuration pathway to produce cystathionine for downstream glutathione biosynthesis in a reaction catalysed by B_6_-dependent cystathionine β-synthase (CBS). (**b**) The structure of SAM and SAH. The strong electrophilic property of the sulfur-linked methyl group in SAM is exploited by methyltransferases to facilitate the methylation of a variety of nucleophile-containing substrates, thereby forming SAH as a product. While methyl group transfer is the most-studied group donation, SAM is also a source of other groups, such as amino, aminoalkyl, and ribosyl moieties [11].

**Figure 2 metabolites-12-00373-f002:**
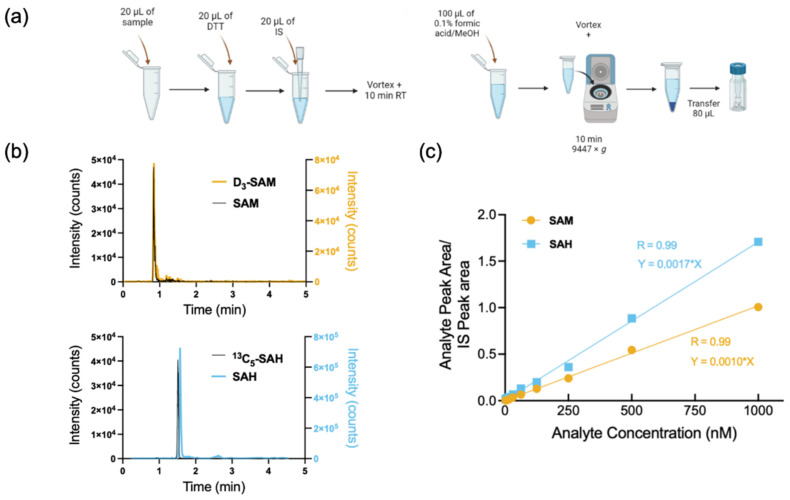
Overview of sample preparation, chromatographic separation, and the linearity of SAM and SAH calibration curves. (**a**) Sample preparation scheme showing 20 µL of starting material (plasma, urine, conditioned culture medium, or cell lysate) extracted into a total volume of 160 µL, followed by centrifugation to spin down proteins and other precipitated biomolecules and transfer of the supernatant into an HPLC vial. (**b**) Chromatograms of SAM and SAH and their respective stable isotopic versions, D_3_-SAM and ^13^C_5_-SAH. (**c**) Calibration curves of SAM and SAH.

**Figure 3 metabolites-12-00373-f003:**
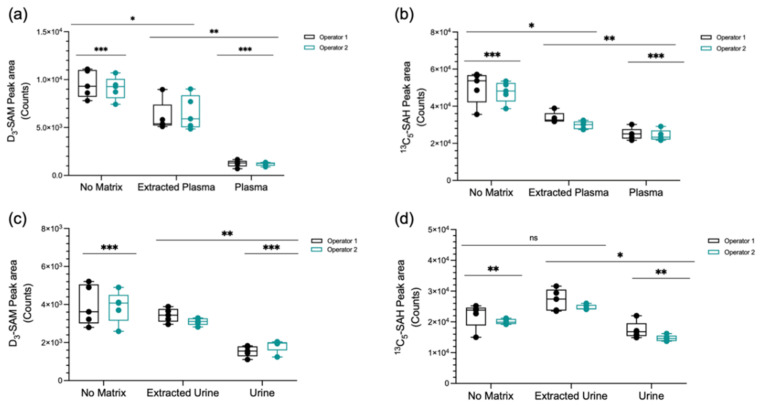
Analysis of the matrix effect by two independent users (operators 1 (black) and 2 (teal)). (**a**) Effect of extracted plasma and native plasma on the peak area of D_3_-SAM. (**b**) Effect of extracted plasma and native plasma on the peak area of ^13^C_5_-SAH. (**c**) Effect of extracted urine and native urine on the peak area of D_3_-SAM. (**d**) Effect of extracted urine and native urine on the peak area of ^13^C_5_-SAH. * *p* ≤ 0.05; ** *p* ≤ 0.01; *** *p* < 0.001; ns: *p* > 0.05.

**Figure 4 metabolites-12-00373-f004:**
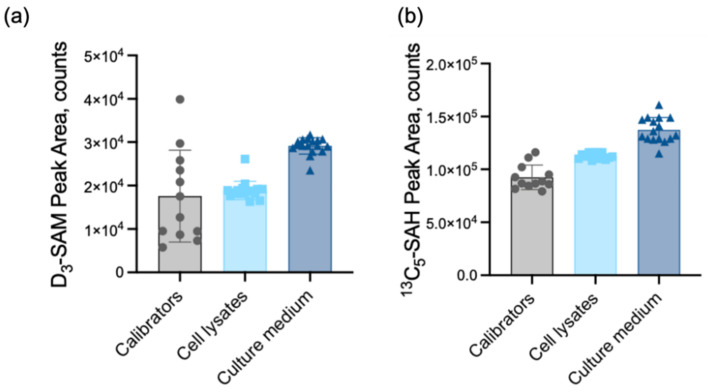
Analysis of the matrix effect for studies with cultured cells. (**a**) Comparison of peak areas of D_3_-SAM in the absence of matrix (calibrators, prepared in 0.1% formic acid in water), with extracted cell lysate or culture medium. (**b**) Comparison of peak areas of ^13^C_5_-SAH in the absence of matrix (calibrators, prepared in 0.1% formic acid in water), with extracted cell lysate or culture medium. Circles: Pure calibrators; Squares: Cell lysate as matrix; Triangles: Culture Medium as matrix.

**Figure 5 metabolites-12-00373-f005:**
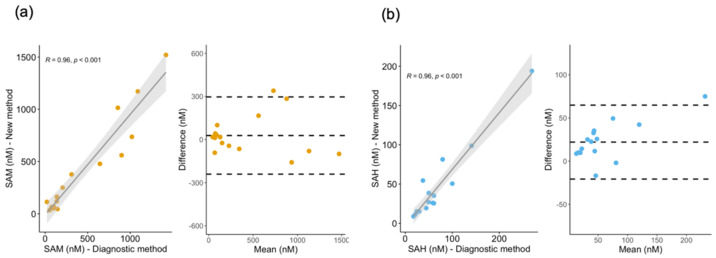
Correlation and Bland–Altman plots of SAM and SAH determinations in plasma by the new method versus an existing diagnostic method. (**a**) Correlation and Bland–Altman plots for the determination of plasma SAM (yellow circles). (**b**) Correlation and Bland–Altman plots for the determination of plasma SAH (blue circles). Grey shadows represent the 95% confidence interval for the linear regression in correlation plots. The middle, dashed line depicts the degree of bias with respect to an ideal 100% accuracy. Differences in concentration values determined by the two methods are predominantly within two standard deviations (± 1.96 SD), herein represented by the top and bottom dashed lines.

**Figure 6 metabolites-12-00373-f006:**
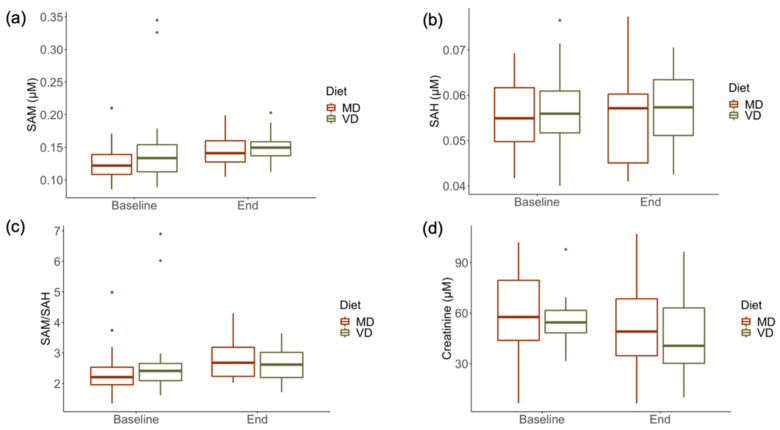
Plasma concentrations of SAM, SAH, the SAM:SAH ratio (SAM/SAH), and creatinine in an RCT with a 4 week intervention with vegan (VD) and meat-rich (MD) diets. (**a**) Plasma SAM at baseline and at the end of the trial. (**b**) Plasma SAH at baseline and at the end of the trial. (**c**) Ratio of plasma SAM and SAH at baseline and the end of the trial. (**d**) Plasma creatinine at baseline and at the end of the trial. Box plots show the median values (lines within the boxes) and the whiskers represent 1.5 interquartile ranges. Values that are classified as outliers are also shown.

**Figure 7 metabolites-12-00373-f007:**
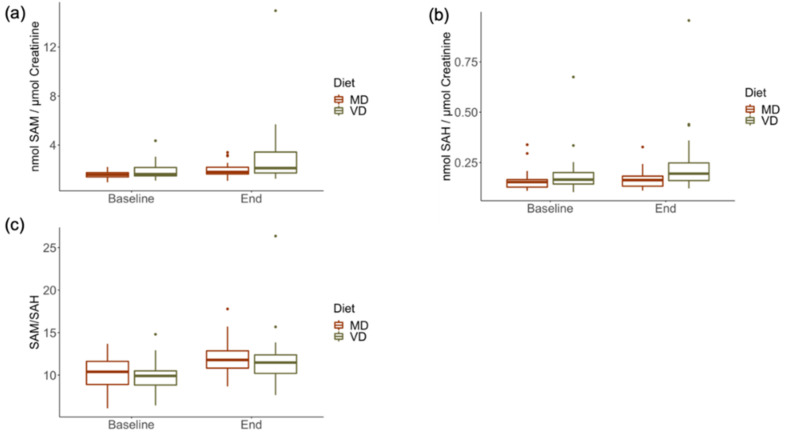
Urinary concentration of SAM, SAH, and the SAM:SAH ratio (SAM/SAH) in an RCT with a 4 week intervention with vegan (VD) and meat-rich (MD) diets. (**a**) Urinary SAM at baseline and the end of the trial. (**b**) Urinary SAH at baseline and the end of the trial. (**c**) Ratio of urinary SAM and SAH at baseline and the end of the trial. The concentrations of SAM and SAH were normalised by the concentration of creatinine to account for varying hydration at the time of sample collection and renal function. Box plots show the median values (lines within the boxes) and the whiskers represent 1.5 interquartile ranges. Values that are classified as outliers are also shown.

**Figure 8 metabolites-12-00373-f008:**
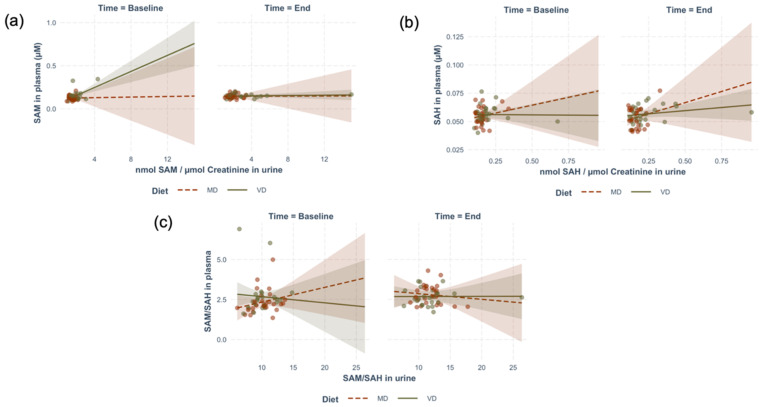
Multiple linear regression analysis of systemic (plasma) and excreted (urinary) SAM and SAH and their SAM:SAH ratios. (**a**) Plasma SAM versus urinary SAM/creatinine at baseline and at the end of the trial. (**b**) Plasma SAH versus urinary SAH/creatinine at baseline and at the end of the trial. (**c**) Plasma SAM:SAH versus urinary SAM/SAH at baseline and at the end of the trial. The shadowed areas represent 95% confidence intervals for the respective linear regressions.

**Table 1 metabolites-12-00373-t001:** Mass transitions and optimised parameters for SAM and SAH.

Analyte	Precursor Ion (Da)	Product Ion (Da)	Dwell Time (msec)	RT (min)	DP (V)	EP (V)	CE (V)	CXP (V)
SAM	399.1	136.1	20	0.83	81	10	37	22
D_3_-SAM	402.1	136.1	20	0.83	51	10	32	14
SAH	385.1	136.1	20	1.53	46	10	23	12
^13^C_5_-SAH	390.1	136.1	20	1.53	11	10	23	6

RT: retention time; DP: declustering potential; EP: entrance potential; CE: collision energy; CXP: collision cell exit potential.

**Table 2 metabolites-12-00373-t002:** Concentration of plasma SAM, SAH, and creatinine in healthy adults (*n* = 33).

	SAM (nM)	SAH (nM)	Creatinine (µM) ^a^	SAM/SAH
Mean ± standard deviation	120.6 ± 18.1	21.5 ± 3.3	42.8 ± 14.9	5.6 ± 1.0
Median (1st quartile–3rd quartile)	124.0 (106.0–134.0)	20.9 (19.5–22.9)	43.2 (32.5–53.8)	6.0 (4.8–6.5)

^a^ Reference ranges of plasma creatinine varies with sex and age [43].

**Table 3 metabolites-12-00373-t003:** Concentration of plasma SAM, SAH, and creatinine in healthy adult subjects at the start of the trial (baseline, omnivorous diet). Data are shown as median (1st quartile–3rd quartile).

Specimen *	Plasma	Urine
Number of subjects	*n* = 53	*n* = 51
SAM (µM)	0.125 (0.112–0.151)	0.832 (0.504–1.160)
SAH (µM)	0.055 (0.051–0.061)	0.079 (0.057–0.109)
Creatinine (µM)	55.5 (47.0–69.4)	509.0 (323.0–774.5)
SAM/SAH	2.3 (2.0–2.7)	10.0 (8.8–11.3)
SAM/Creatinine (nmol/µmol)	-	1.609 (1.434–1.906)
SAH/Creatinine (nmol/µmol)	-	0.156 (0.134–0.183)

* Plasma samples were collected for all 53 participants, whereas urine samples were collected for 51 of the 53 participants (subjects 19 and 21, both females, did not provide urine samples).

**Table 4 metabolites-12-00373-t004:** Review of published SAM and SAH concentrations in human plasma.

Study	Lind, M. V. et al. [18]	Struys, E. A. et al. [42] (*)	Lind, M. V. et al. [44]	Melnik, S. et al. [45]	Kirsch, S. H. et al. [41]	Klepacki, J. et al. [46]	James, S.J. et al. [47]	Stabler, S. and Allen, R. [48]	Guiraud, S.P. et al. [49]	This Study
**SAM (nM)**	89.7 ± 15.3	104 ± 4.4	90.2 ± 14.5	73.5 ± 6.7	83.6 ± 7.8	95.2 ± 21.6	96.9 ± 12	109 (71–168)	88.5 ± 18.1	120 ± 18
**SAH (nM)**	16.5 ± 6.5	46.5 ± 1.9	16.3 ± 5.6	22.6 ± 1.3	12.7 ± 1.5	30.4 ± 6.2	19.4 ± 3.4	15 (8–26)	25.7 ± 9.9	21.5 ± 3.3
**SAM/SAH**	5.9	2.2	5.9	3.2	6.6	3.1	5.2	7.4	3.4	5.6
**N**	118, 47 men and 71 women	15 women	118, 47 men and 71 women	58 healthy women	31 healthy individuals of which 6 are men	8 healthy volunteers (47 de novo kidney transplant patients)	33 healthy children	48 healthy controls	6 healthy controls	33 adult healthy controls
**Method**	LC-MS/MS	LC-MS/MS	LC-MS/MS,	HPLC with coulometric detection	UPLC-MS/MS	LC-MS/MS	HPLC with electrochemical detection	LC-MS/MS	LC-MS/MS	LC-MS/MS
**Amount of sample**	100 µL plasma	500 µL plasma	100 µL plasma	200 µL plasma	500 µL plasma	200 µL plasma	200 µL plasma	500 µL plasma, serum or CSF 50 µL urine	50 µL plasma or CSF	20 µL plasma (or 20 µL urine, cell lysate, culture medium)
**Comments**	Average age 49 years old, overweight, slightly elevated metabolic features	Pre-menopausal women with normal Hcy	Average age 49 years old, overweight, slightly elevated metabolic features	Healthy adult females, mean age 37.2 years old	Developed by diagnostic laboratory in Germany	FDA validated study USA-Germany cooperation	Children aged 7.4 ± 1.3 years old	Recommended storage: −80 °C and thawing in ice on the day of the assay	SAM and SAH along with other metabolites by SRM	QC: internal. Commercial human plasma spiked with SAM and SAH
**Sample cleanup**	Acidic extraction Neutralisation SPE	Acidic extraction Neutralisation SPE	Acidic extraction Neutralisation SPE	Acidic extraction 0.22 µm filtration	Neutral pH SPE	Acetone extraction (only IS in acidic solution)	Reduction and acidic extraction	Acidic extraction SPE	Reduction Acidic extraction 0.22 µm filtration	Acidic extraction

Abbreviations: LC-MS/MS: liquid chromatography tandem mass spectrometry; SPE: solid phase extraction; IS: internal standard; SRM: single reaction monitoring; FDA: Food and Drug Administration; N: Number of participants. (*) Method utilized by the diagnostic laboratory for cross-comparison of measurements performed in this study.

**Table 5 metabolites-12-00373-t005:** Review of published SAM and SAH concentrations in human urine.

Study	Stabler, S. and Allen, R. [48]	Ivanov, A.V. et al. [50]	Ivanov, A.V. et al. [51]	Kruglova, M. P. et al. [52]	This Study
**SAM (µM)**	19.1–124.8	12.1	30–60	10.2	22.1 ± 11.0
**SAH (µM)**	0.08–3.1	0.73	6–30	0.89	2.2 ± 1.0
**SAM/SAH**	4.5–94.4	16.58	2–5	11	10.0 ± 1.9 (6.1–14.8)
**N**	48 (only *n* = 6 for urine)	30 urine samples from healthy volunteers	40 healthy volunteers (20–75 years old)	20 healthy controls	53 adult healthy controls (18–60 years old)
**Method**	Stable isotope dilution LC-MS (SIM)	Micellar electrokinetic chromatography (MEKC/CE)	CE/UV detection	CE/UV detection	LC-MS/MS
**Amount of sample**	50 µL of urine	300 µL of urine	2000 µL	2200 µL of urine	20 µL urine 1:25
**Comments**	Recommended storage: −80 °C and thawing in ice on the day of the assay	Median SAM/Creatinine (µM/mM): 1.17 Median SAH/Creatinine (µM/mM): 0.083	LOD SAM and SAH: 0.07 µmol/L (S/N = 3) LOQ SAM and SAH: 0.2 µmol/L (S/N = 10) Linearity SAM: 0.06-40 µM Linerity SAH: 0.09-7 µM	Mean age of 58.5 years (range, 46–71 years; 43.5% male) SAM/Creatinine 2.33(µmol/mmol) SAH/Creatinine 0.201 (µmol/mmol)	SAM/Creatinine (nmol/µmol): 1.609 (1.434–1.906) SAH/Creatinine (nmol/µmol): 0.156 (0.134–0.183) SAM/SAH: 10.0 (8.8–11.3)
**Sample Cleanup**	Acidic extraction, SPE	SPE	SPE	SPE	Acidic extraction

Abbreviations: SPE: solid phase extraction; CE: capillary electrophoresis; SIM: selected ion monitoring; UV: ultraviolet detection; S/N: signal to noise ratio.

**Table 6 metabolites-12-00373-t006:** Effect of dietary intervention on the concentrations of SAM and SAH in the plasma and urine of healthy subjects. Values are expressed as medians (1st quartile–3rd quartile) for descriptive statistics and χ^2^ (*p*-value) from a generalised linear mixed model (GLMM). *p*-values < 0.05 are highlighted in bold font.

**Plasma**					**GLMM for Plasma**		
	**MD**		**VD**		**Fixed Effects and Interactions χ^2^ (*p*-Value)**		
	**Baseline**	**End**	**Baseline**	**End**			
	**(*n* = 27)**	**(*n* = 27)**	**(*n* = 26)**	**(*n* = 26)**	**Time**	**Diet**	**Time × Diet**
**SAM (µM)**	0.122 (0.108–0.139)	0.141 (0.127–0.160)	0.135 (0.112–0.154)	0.149 (0.137–0.158)	**4.363 (0.036)**	2.976 (0.084)	2.029 (0.154)
**SAH (µM)**	0.054 (0.049–0.061)	0.057 (0.045–0.060)	0.055 (0.051–0.060)	0.057 (0.051–0.063)	0.073 (0.786)	1.463 (0.226)	1.129 (0.287)
**Creatinine (µM)**	57.7 (43.9–79.5)	49.1 (34.8–68.5)	54.5 (48.3–61.7)	40.7 (30.2–63.1)	**5.009 (0.025)**	1.790 (0.180)	0.0009 (0.975)
**SAM/SAH**	2.2 (2.0–2.5)	2.7 (2.2–3.2)	2.4 (2.1–2.7)	2.6 (2.2–3.0)	**4.692 (0.030)**	0.269 (0.603)	2.996 (0.083)
**Urine**					**GLMM for Urine**		
	**MD**		**VD**		**Fixed Effects and Interactions** **χ^2^ (*p*-Value)**		
	**Baseline**	**End**	**Baseline**	**End**			
	**(*n* = 26)**	**(*n* = 26)**	**(*n* = 25)**	**(*n* = 25)**	**Time**	**Diet**	**Time × Diet**
**SAM/Creatinine (nmol/µmol)**	1.569 (1.394–1.738)	1.792 (1.626–2.180)	1.627 (1.483–2.161)	2.113 (1.716–3.423)	**26.976 (< 0.001)**	**7.101 (0.007)**	2.725 (0.098)
**SAH/Creatinine (nmol/µmol)**	0.153 (0.128–0.165)	0.163 (0.132–0.182)	0.165 (0.143–0.200)	0.195 (0.160–0.248)	**5.356 (0.020)**	**8.025 (0.004)**	2.387 (0.122)
**SAM/SAH**	10.4 (8.9–11.6)	11.8 (10.8–12.9)	9.9 (8.8–10.5)	11.5 (10.2–12.4)	**23.687 (< 0.001)**	0.374 (0.540)	0.024 (0.8761)

**Table 7 metabolites-12-00373-t007:** Results of the multiple linear regression analysis of SAM, SAH, and SAM/SAH in plasma and urine. Entries highlighted in bold font are statistically significant (*p*-value < 0.05).

	SAM in Plasma		SAH in Plasma		SAM/SAH in Plasma	
	**F-Value**	***p*-Value**	**F-Value**	***p*-Value**	**F-Value**	***p*-Value**
Metabolite_Urine	0.005	0.939	5.687	**0.019**	0.920	0.339
Time	12.038	**0.0008**	0.467	0.495	11.775	**0.0009**
Diet	0.735	0.393	0.678	0.412	0.000	0.975
Metabolite_Urine × Time	0.53	0.468	0.220	0.639	2.162	0.144
Metabolite_Urine × Diet	0.002	0.961	0.478	0.491	0.761	0.385
Time × Diet	0.022	0.880	0.055	0.814	1.608	0.20
Metabolite_Urine × Time × Diet	0.154	0.695	2.506	0.116	0.002	0.959

## Data Availability

The data are available via the Appendix A. Further information is available from the corresponding author upon reasonable request.

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
