# Peer review of "Analysis of S-Adenosylmethionine and S-Adenosylhomocysteine: Method Optimisation and Profiling in Healthy Adults upon Short-Term Dietary Intervention"

_metabolites, 2022, doi:10.3390/metabo12050373_

Round 1

Reviewer 1 Report

This study is very interesting assessment to explore the SAM and SAH concentration in plasma requires freshly isolated plasma 357 and rapid freezing, and ideally acidification of samples prior to long-term storage and 358 measurement. 

however, there are some points need to be addressed and clarified the results.

1, In the Figure 3, the comparsion between two operators did not clear present.  the difference labelling made it hard to understanding. 

2, How to explain the difference results in the Figure 4?  

Reviewer 2 Report

Review of Metabolites 1639703-v1

General comments: 

This paper is far from my area of expertise.  I accepted the review because the abstract mentioned the diet study.  If my comments are off-base, feel free to ignore them.

I gather that there is no “gold standard” for the analysis of these chemicals in plasma and urine, and that the method proposed in this paper has several advantages over other commonly used assays, namely small sample size, fewer steps, shorter elapsed time, common availability of the necessary reagents.

I am somewhat surprised that it is acceptable to determine reference ranges based on a small (n=33) sample, and that the way of comparing various assays is to see whether the reference ranges are similar.

In much of the paper, the use of median (quartiles) for reporting means that it is not clear what the very high or very low numbers mean.

Specific comments on Table 4, particularly the last 3 columns:

It took a while to figure out what is being presented.  Perhaps using column headers, rather than placing all the information in the table title would make it more comprehensible.  I am accustomed to the presentation of regression coefficients with standard errors, whereas here I (think I) see only point estimates with p-values.  Given that some of the models do not have identity link, it is not clear to me what is being presented, and on what scale, since the numbers in the last 3 columns are an order of magnitude greater than those presented in the first 4 columns.

Outliers:

In figures 6 and 7 (and potentially in figure 8, except that they were excluded) there are substantial outliers.  Are these related to the subjects (i.e., some true reaction to the VD on the part of some people) or to the assays?  If the authors are proposing their assay for clinical use, this needs exploration, as the authors note in 2.6.3.  I gather that the authors only did their own test during this study, not the local diagnostic test.

Figure 8

Essentially, for each of the 3 comparisons, 4 models are made (implicitly).  It seems to me that for each comparison, they could make a unified model

~ diet time diet*time

              * diet    * time    *diet*time

And check whether the last 3 interactions make a difference (e.g., by some sort of score test).

Another possibility would be to use some sort of data augmentation method that would allow a contrast of the first coefficient in the 2 models.

Getting into something I know, the trial would have been much stronger as a crossover (with a washout omnivore period between), assuming that the purpose of the trial was not to do this analysis, but to study some other issue(s), but perhaps that is too much to ask of subjects.

Round 2

Reviewer 2 Report

The analysis and the presentation have been improved.

I have no further comments on this paper, except to note that i counted 42 explicit statistical tests in the tables with 9 p-values < .05.  The authors might wish to mention multiple testing.